# SwimBIT: A Novel Approach to Stroke Analysis During Swim Training Based on Attitude and Heading Reference System (AHRS)

**DOI:** 10.3390/sports7110238

**Published:** 2019-11-16

**Authors:** Eduardo Ramos Félix, Hugo Plácido da Silva, Bjørn Harald Olstad, Jan Cabri, Paulo Lobato Correia

**Affiliations:** 1Instituto Superior Técnico, Universidade de Lisboa, 1049-001 Lisboa, Portugal; eduardo.felix@ist.utl.pt (E.R.F.); plc@lx.it.pt (P.L.C.); 2IT—Instituto de Telecomunicações, 1049-001 Lisboa, Portugal; 3Department of Physical Performance, Norwegian School of Sport Sciences, 0863 Oslo, Norway; b.h.olstad@nih.no; 4Luxembourg Institute of Research in Orthopedics, Sports Medicine and Science, 1460 Luxembourg, Luxembourg; jan.cabri@liroms.lu

**Keywords:** swimming, training, performance, swimming analysis, inertial measurement units (IMU)

## Abstract

In a world where technology is assuming a pervasive role, sports sciences are also increasingly exploiting the possibilities opened by advanced sensors and intelligent algorithms. This paper focuses on the development of a convenient, practical, and low-cost system, SwimBIT, which is intended to help swimmers and coaches in performance evaluation, improvement, and injury reduction. Real-world data were collected from 13 triathletes (age 20.8 ± 3.5 years, height 173.7 ± 5.3 cm, and weight 63.5 ± 6.3 kg) with different skill levels in performing the four competitive styles of swimming in order to develop a representative database and allow assessment of the system’s performance in swimming conditions. The hardware collects a set of signals from swimmers based on an attitude and heading reference system (AHRS), and a machine learning workflow for data analysis is used to extract a selection of indicators that allows analysis of a swimmer’s performance. Based on the AHRS data, three novel indicators are proposed: trunk elevation, body balance, and body rotation. Experimental evaluation has shown promising results, with a 100% accuracy in swim lap segmentation, a precision of 100% in the recognition of backstroke, and a precision of 89.60% in the three remaining swimming techniques (butterfly, breaststroke, and front crawl). The performance indicators proposed here provide valuable information for both swimmers and coaches in their quest for enhancing performance and preventing injuries.

## 1. Introduction

Video-based and inertial sensing approaches (using wearable devices) are often used to support swimmers and coaches in their daily work for improving performance and preventing injuries. Among the two, video-based approaches are most widely used. Even a simple setup, for example, using a mobile phone (out of water) or an action cam (underwater), enables recording of swimmer movements for further or real-time analysis. Cameras can also be waterproofed to obtain underwater images [1], a transparent underwater window can be used in specially prepared pools [2], or a periscope system can be employed [3]. Video-based approaches have limitations as they often require visual markers on the swimmer, which may hinder the swimmer in executing movements in ecological conditions. These markers might introduce added drag and limit the swimmer’s movements, which may lead to an observation that differs from the normal swimming. Furthermore, camera calibration is difficult, and the image is not stabilized. The swimmer’s movements also disturb visibility, which can hinder the acquisition of good-quality images, and often a single viewpoint of the swimmer is captured. However, more complex setups can be considered, as the ones proposed by different manufacturers [4,5,6] or similar systems, although these have high installation and running costs. Some systems also consider multiple recording cameras that require larger technical requirements for synchronization. Data processing requires the markers in each video frame to be located in order to reconstruct the swimmer’s movement and extract relevant parameters. Marker visibility is often impaired by water turbulence, refraction, or reflections of light.

Inertial sensing approaches use waterproof inertial measurement units (IMUs) and have several advantages. Firstly, there is no need to locate the markers in noisy images. Secondly, wearing a small device causes less discomfort than a complete swimsuit with visual markers attached to it (or wearing the markers on the skin). Thirdly, the IMUs can store large amounts of information and process parts of it, so results can be readily accessed once transferred to a computer. Fourthly, IMUs enable simultaneous assessment of many/all swimmers and are not limited to one at a time. Lastly, this is an affordable solution, and every swimmer can easily access it. However, during competitions, setting up swimmers with sensors is still not allowed, and their usage is therefore limited to only training sessions.

This work explores the use of an attitude and heading reference system (AHRS), which uses a wearable device together with signal processing and machine learning workflows, for technical gesture classification and computation of selected performance indicators. To overcome the abovementioned limitations and further help coaches and swimmers in their daily training, the developed prototype collects inertial signals from a swimmer with the purpose of optimizing technical gestures and reducing the risk of injuries associated with technical stroke errors. Furthermore, trunk elevation, body balance, and body rotation, derived from AHRS data, are introduced as novel performance indicators that extend the conventional set of indicators.

## 2. Materials and Methods

### 2.1. Hardware

The hardware of the SwimBIT device includes an IMU with a 3-axis accelerometer, gyroscope, and magnetometer, allowing calculation of the absolute orientation of the module in space [7]. By default, the device streams the acquired data via WiFi. However, as signals are strongly attenuated in the aquatic environment [8], leading to frequent connection losses, the device was modified to store data directly on a memory card [9]. Data logging is done at 100 Hz, and the following parameters are recorded in comma-separated values (CSV) format: sampling time (ms), nine channels of the accelerometer (g), gyroscope (deg.s^−1^), magnetometer (gauss), and three Euler and heading angles (°). The developed firmware is publicly available as open source [10].

Figure 1a shows the developed prototype and the axes used by the IMU. An important aspect is the placement of the device on the swimmer’s body. Pansiot et al. [11] demonstrated that the trunk or head locations allow the collection of reliable data about lap count and time, overall momentum, stroke count, body roll, and breathing patterns. Placing the IMU on the arms or the legs also provides the possibility to measure symmetry of arm strokes or kicks. It was therefore decided that the device would be fixed, inside a sealed bag, on the swimmer’s lower back, as illustrated in Figure 1b. The swimmers felt comfortable with the presence of the device, favoring it over placements on the wrist, head cap, or goggles. Overall, the hardware is 53 × 29 × 5 mm in size and weighs 15 grams. When placed in the sealed bag, the total dimensions of the arrangement are 110 × 105 × 8.5 mm, and the weight increases to 55 grams.

### 2.2. Procedures

Data were collected in a 25 meter pool from 13 triathletes (11 males and 2 females) between 18 and 29 years old (with an average age of 20.8 ± 3.5 years), height between 164 and 182 cm (with an average of 173.7 ± 5.3 cm), weight between 55 and 74 kg (with an average of 63.5 ± 6.3 kg), and different expertise levels (4 international, 6 national, and 3 retired). Each athlete was asked to swim 100 meters in each technique, resulting in a total of 400 meters per participant, not specifying the order of swimming techniques or the number of stops made during the data acquisition. 

An example of the acquired pitch signal is included in Figure 1c. One swimmer swam only 50 meters butterfly, another swam 50 meters of each stroke style, and one swam 100 meters of front crawl at the end. A total of 202 laps were obtained: 48 butterfly, 50 backstroke, 50 breaststroke, and 54 front crawl. Protocols were submitted to and approved by the institutional ethics committee.

Data processing algorithms were developed to analyze the swimmer’s performance following the workflow represented in Figure 2. The initialization block retrieves parameters such as the length of the pool and metadata about the swimmer, among others. Afterward, the CSV file containing the acquired data was loaded, and the desired performance indicators were computed.

### 2.3. Performance Indicators

The swimming performance indicators computed by the SwimBIT system include the turns and stops, stroke type estimation, number of strokes in each lap, body balance and rotation statistics, trunk elevation, stroke rate, and lap times as follows:turns and stops: the beginning and ending time as well as the duration of each turn and stop;stroke technique: the stroke style swam by the swimmer in each lap;lap time: each lap time in seconds;stroke count: the number of strokes in each lap;stroke rate: the stroke rate in strokes per minute per lap;trunk elevation: the elevation angle of the trunk in each stroke;body balance: evaluation of the range of angular body motions depending on the style; andbody rotation: evaluation of the body roll angle.

### 2.4. Turns and Stops

Identifying turns and stops allows each lap to be segmented, after which the swimming-specific features can be extracted. As the swimmer is in an approximately horizontal position during the four styles, the pitch angle can be analyzed to detect turns and stops. While swimming, the pitch typically takes values lower than ±20°, as illustrated in Figure 1c. When the swimmer makes a turn (either a tumble or an open turn), the pitch angle rises to a high value (close to ±90°). In a stop, the pitch angle changes from nearly 0° to −90° (when the swimmer is standing). A stop is considered if the separation between the local maximum and minimum is at least 3 seconds.

Figure 1c shows the detected turns (blue circles) and stops (blue crosses) after analyzing the pitch signal.

In case the swimmer performs a bucket turn (backward flip starting with the swimmer’s palm touching the wall) or a cross-over turn (the swimmer touches the wall with their upper arm crossing over the body, and the back is driven in the touching hand direction to complete the turn), the pitch angle may not detect the turn for the backstroke style. These cases are detected by analyzing the pattern of lap time duration, and excessively long “lap times” are identified as outliers by analyzing the median absolute deviation (*MAD*), where A is a random variable vector, composed up of N scalar observations (Equation (1)). An observation is then labelinged as an outlier when its value is more than three scaled median absolute deviations away from the median.
(1)$$MAD=median\left(\left|A_{i}-median\left(A\right)\right|\right),i=1,2,\ldots ,N$$

The turn is determined when the heading angles reverse and the pitch has a maximum value. As the time spent by the swimmer in turns can greatly impact swimming performance, the proposed algorithm also determines the precise instants when the turns/stops begin and end. This is done by sliding a window of 0.5 seconds before and after the detected turn/stop instant and searching for the moment when the window average value is in the normal swimming range of ±20°. Examples of detected beginning and ending of turns/stops are shown in Figure 1c.

### 2.5. Stroke Technique

Having identified each lap (between turns and stops), the stroke style can be identified so that further performance indicators can be computed. The SwimBIT system implements a technique based on an algorithm proposed by [13]. It first filters the acceleration signal to remove the high-frequency components, determines the orientation of the swimmer, and applies a classifier to determine the stroke style.

Filtering of the acceleration signal is done using a 48-order Hamming low-pass filter with a cutoff frequency of 0.5 Hz [14]. The vertical axis (z-axis) of the accelerometer signal is then analyzed to determine if the style is backstroke, the only style where the swimmer’s back is turned toward the pool bottom (inverted z-axis). To distinguish the other stroke styles, the energy of each accelerometer channel is computed according to Equation (2).
(2)$$E_{channel}\;=\;round\left(\frac{\sum_{n=1}^{N}\left|x\left(n\right)-\bar{x}\right|}{N}\right)$$

Front crawl can be identified as having much higher energy in the y-axis channel due to the rotation of the swimmer’s body in each stroke. Breaststroke and butterfly techniques can be distinguished as butterfly shows higher energy in the z-axis due to the stroke’s undulatory movement as well as in the x-axis because it is generally a faster style than breaststroke (although not for everyone).

Representing the average energy of each accelerometer channel in each lap allows the training of a classifier to automatically distinguish the front crawl, butterfly, and breaststroke styles. In this work, two supervised machine learning approaches were tested: a support vector machine (SVM) and an artificial neural network (ANN); these are compared in the results section. SVMs are an efficient kernel-based method for linear and nonlinear classification, which implicitly map the input data into high-dimensional feature spaces with the goal of maximizing the interclass separation. ANNs consist of a multilayer network arrangement of individual processing functions that take features or signals as inputs. In their simplest form, the network consists of an input layer (fed with the features), a hidden layer, and an output layer (that provides the class labels). Each connection between nodes has an associated weight, and the network adjusts those weights during training to maximize the mapping between the input features and the class labels. For more detailed information about these methods, we refer the reader to [15,16].

### 2.6. Stroke Count

The number of strokes in a lap is computed differently depending on the swimming style. For nonsymmetrical styles (front crawl and backstroke), each arm stroke is accompanied by a trunk rotation. Therefore, the swimmer’s lower back rotation angle (roll) exhibits an approximately sinusoidal wave, where each peak (both maxima and minima) represents a stroke, as illustrated in Figure 1d. Stroke counting can be performed by filtering the roll signal using a 48-order low-pass Hamming filter with a cutoff frequency of 3 Hz and searching for local maxima and minima. Additional restrictions are imposed to avoid false detections, such as a minimum body rotation of 20° (with this value being determined based on numerical analysis of the overall patterns in the database).

For symmetrical styles (breaststroke and butterfly), stroke counting relies on the pitch angle analysis. Again, the signal is filtered using a 48-order low-pass Hamming filter with a cutoff frequency of 3 Hz and local maxima and minima search. A stroke cycle corresponds to the segment between consecutive minima. 

To remove outlier detections, dynamic time warping (DTW) is used. All participants’ stroke cycle segments are resampled to contain 200 samples, and a mean wave is created based on the central 50% of the pitch signal (from 25% to 75% of the signal between consecutive minima). Then, DTW is used to align each participant’s stroke signal and the mean wave, minimizing the Euclidean distance between them. If the distance between the participant’s segment and the mean wave is larger than an empirically determined threshold, the participant’s segment is discarded so that only true stroke cycles are counted.

For both symmetrical and nonsymmetrical styles, the counting is based on the detection of the pitch signal minima or the roll signal minima and maxima. A threshold operation is performed according to Equation (3), where each sample is analyzed. $A_{i}$, from a vector A, is compared against its mean value, $\bar{A}$, plus k times its standard deviation, $\sigma_{A}$.
(3)$$\bar{A}-\left(k\cdot \sigma_{A}\right)<A_{i}<\bar{A}+\left(k\cdot \sigma_{A}\right)$$

After initial tests, the values of k were selected as k=2 for front crawl, k=2.36 for backstroke, k=0.74 for breaststroke, and k=1.34 for butterfly. 

### 2.7. Trunk Elevation

For symmetrical strokes (breaststroke and butterfly), the trunk elevation is computed as the difference between the minimum and maximum pitch angles within a stroke cycle; in our proposed approach, the average trunk elevation per lap is also computed.

### 2.8. Body Balance

The body balance is evaluated by computing the pitch angle. There are two ways to present this indicator, depending on the type of style the swimmer is performing. In the case of nonsymmetrical styles (front crawl and backstroke), the goal is to keep a pitch constant and close to 0°. In this way, the average lap pitch is computed. On the other hand, in the case of symmetrical styles (breaststroke and butterfly), the pitch varies greatly within each stroke cycle. Thus, for all given strokes in a lap, it is possible to analyze the minimum and maximum pitch angles. In each lap, the maximum and minimum pitch averages are also computed.

### 2.9. Body Rotation

The body rotation is evaluated by computing the roll angle. This feature is presented in an analogous way to the previous one, changing the symmetrical stroke styles to the nonsymmetrical styles and vice versa. Thus, in the case of symmetrical styles, the average rotation of the swimmer’s body during a given lap is computed (a value that should be close to 0°), while in the case of nonsymmetrical styles, two vectors are returned—one of maxima and one of minima—with information related to the strokes given by the swimmer.

## 3. Results

### 3.1. Turns and Stops

The collected data included a total of 202 laps, each delimited by turn or stop events. All the delimiting events were correctly detected by the proposed algorithms, leading to 100% accuracy in lap segmentation. All stops were correctly detected, but two out of the 181 turns in the dataset were incorrectly detected as stops. Ground truth for these assessments was collected using a combination of video analysis and manual observation.

The two misclassifications corresponded to two consecutive turns that took more than three seconds to complete by the same swimmer (an unexpected value in a competitive context). As an adjustable parameter is used to specify the time limit to distinguish a turn from a stop, a coach can easily adjust it to suit the expertise level of the swimmers being monitored.

### 3.2. Stroke Technique

The swim style adopted in each lap was estimated by analyzing the z-axis accelerometer value to determine the backstroke technique (if the value is around −1 g). Two different automatic classifiers (SVM and ANN) were tested to distinguish the other three techniques. In this work, the dataset consisted of 48 (butterfly) + 50 (breaststroke) + 54 (front crawl) = 152 laps, of which 122 laps were randomly selected for training and the remaining 30 laps were used for testing, adding up to a total of 1000 runs performed.

The front crawl, breaststroke, and butterfly laps of the dataset are represented in Figure 3. It can be seen that the dataset is not linearly separable, notably in terms of the butterfly and breaststroke styles. 

For the SVM classifier, the best accuracy was obtained using a polynomial kernel with soft margin $BoxConstraint=10^{5},\rho =1$. The ANN included three layers: the input layer composed of three neurons (corresponding to energy values of the three accelerometer axes), a hidden layer composed of 10 neurons, and the output layer composed of three neurons (corresponding to the three stroke styles: butterfly, breaststroke, and front crawl). The best performance was obtained when training with the gradient descent with momentum algorithm (with constant α=0.5 and learning rate η=0.3).

The backstroke style was recognized with 100% accuracy in the 50 laps contained in the dataset. For the other three stroke styles, the best results obtained using the two classifiers were very similar; the accuracy and standard deviation values were 89% ± 5.51 for the SVM and 89.6% ± 5.67 for the ANN.

### 3.3. Stroke Count 

Performance assessment of our stroke counting module was based on accuracy defined in Equation (4), where FP represents the false positives, FN the false negatives, TP the true positives, and TN the true negatives, which can then be used for performance assessment. A FP is a stroke not accounted for by the algorithm, while a TN is a stroke candidate that is correctly rejected.
(4)$$Accuracy=\frac{TP+TN}{TP+FP+TN+FN}$$

The obtained accuracy results were as follows: 91.52% for breaststroke, 91.99% for butterfly, 97.48% for front crawl, and 95.69% for backstroke. Some of the observed problems correspond to the last butterfly stroke occasionally not being recognized as it is performed in a continuous rotation of the trunk that culminates in the turn approach. 

### 3.4. Body Balance, Body Rotation, and Trunk Elevation

The novel parameters were analyzed in terms of range of motion per stroke and standard deviation. We studied each style and grouped the athletes by gender as athletic differences between male and female subjects are generally found. A summary of the results is presented in Table 1.

## 4. Discussion

The SwimBIT system consists of low-cost hardware (~€150) as well as signal processing and machine learning workflows to process the generated signals, with the aim of providing a precise, comfortable, and functional device for measurements of swimming technique in order to enhance performance and prevent injuries. This was achieved using an AHRS method (validated by [17]) and adapting it to the aquatic environment with the necessary hardware and firmware changes. Besides providing instantaneous values, the SwimBIT system also computes training session statistics.

The results obtained rely largely on the Euler angles analysis from which the parameters of body balance, body rotation, and trunk elevation are extracted. To the best of the author’s knowledge, these have not been previously combined to compute swimming performance in available existing commercial systems or state-of-the-art research.

The performance indicators proposed here provide valuable information for both swimmers and coaches in their quest for enhancing performance and preventing injuries. The body balance is a pertinent marker for swimming technique and performance assessment, and it is evaluated from computing the pitch angle. In the nonsymmetrical styles (front crawl and backstroke), the goal is to keep a pitch angle constant and close to 0°. In this way, the average lap pitch can be computed. The values associated with the mean and standard deviation are shown in the body balance column of the front crawl and backstroke techniques in Table 1. An increase in pitch angle would indicate a stroke technique where the legs stay deeper in the water and may therefore hamper the body position and lead to an increase in the active drag of the swimmer. An increase in pitch angle can, for example, occur due to poor floatation abilities or from a weak or improper leg kick. In the symmetrical styles (breaststroke and butterfly), the pitch angle will vary greatly within each stroke cycle and is related to the undulating body movement seen in these two styles. It is therefore a temporal event in each stroke cycle, where the swimmer’s trunk reaches its maximum and minimum elevation level. The values associated with the mean and standard deviation for the maximum and minimum elevation levels are shown in the body balance column of the breaststroke and butterfly styles in Table 1. There are also large differences between swimmers in these two styles in relation to body undulation and pitch angle. This parameter will be valuable in order to analyze the stroke cycle to stroke cycle variability of the swimmer and for the coach and swimmer to evaluate whether they should incorporate more or less body undulation into the technique. If a swimmer is performing an excessive amount of body undulation (high maximum value), this could impose additional drag, or if the stroke is performed with too little body undulation (low maximum value), the stroke would be considered too flat on the surface. The minimum value can also be used to determine how horizontal the swimmer’s body is during the gliding phase of the breaststroke.

Body rotation is another important parameter for evaluating swimming technique for enhanced performance and injury prevention. Nonsymmetrical styles (front crawl and backstroke) are performed through a longitudinal axis rotation of the swimmer’s body. As such, there are strokes with rotation to the left and right sides. The mean and standard deviation of the rotation obtained in each side are shown in the body rotation column in Table 1 (maximum to one side and minimum to the other). This is a good performance indicator for revealing asymmetries in the stroke technique as the body rotation should be equal on both sides. This is also interesting in front crawl in terms of investigating what occurs to the body rotation when the swimmer takes a breath (also because some swimmers breathe on either even or odd strokes). Similarly, body rotation can also help to detect structural problems in the swimmer’s technique. For instance, poor body rotation results in a poor reach after the hand/arm entry, reducing the stroke length and thus not utilizing core strength to produce additional force during the arm stroke. In addition, a lack of body rotation imposes a higher demand on the shoulder, which can lead to overuse injuries over time. In contrast, excessive rotation of the torso unbalances and slows down the swimmer. The average of the body rotation in each lap in the symmetrical styles of breaststroke and butterfly (theoretically there is no body rotation) was also calculated. The purpose was to see if the swimmers tended to tilt the body to either side, which can indicate a strength imbalance between the left and right sides or a general lack of balance in the water. The values associated with the mean and standard deviation are presented in the body rotation column of the breaststroke and butterfly techniques in Table 1.

Trunk elevation can provide useful information for the symmetrical styles of breaststroke and butterfly. These styles are performed through an elevation of the swimmer’s body. For example, insufficient trunk elevation leads to a higher water resistance during arm stroke recovery as the arms are more submerged than necessary. An excessive trunk elevation represents unnecessary energy expenditure and is prone to fatigue and slow down the swimmer’s motion.

Future work will focus on testing the SwimBIT placement on other body locations, usage of multiple units, and improvement of the classification accuracy, which constitute limitations to the present study.

## Figures and Tables

**Figure 1 sports-07-00238-f001:**
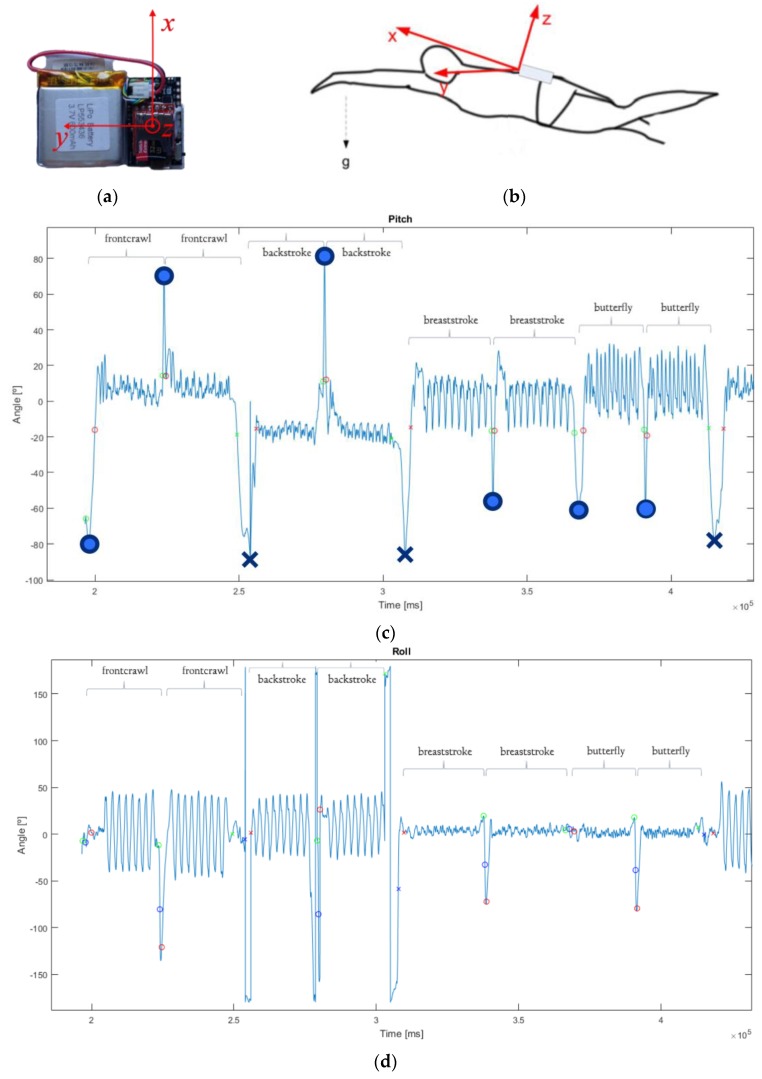
SwimBIT system overview. (**a**) Developed inertial measurement unit (IMU) with data logger and battery. (**b**) Body reference frame for a swimmer (adapted from [12]), where x, y, and z are the forward, side-to-side, and vertical motions of the swimmer, respectively. (**c**,**d**) Examples of acquired data, representing the swimmer’s body pitch and roll, respectively (crosses represent the detected stops, circles represent the detected turns).

**Figure 2 sports-07-00238-f002:**
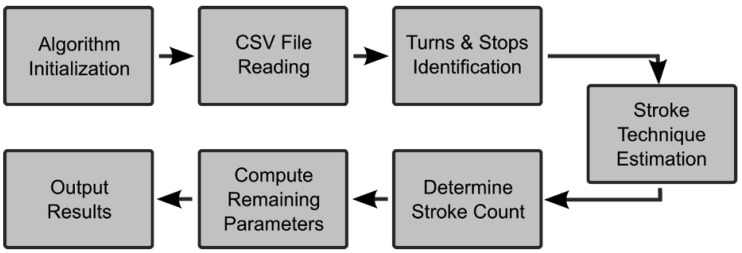
IMU data analysis system workflow.

**Figure 3 sports-07-00238-f003:**
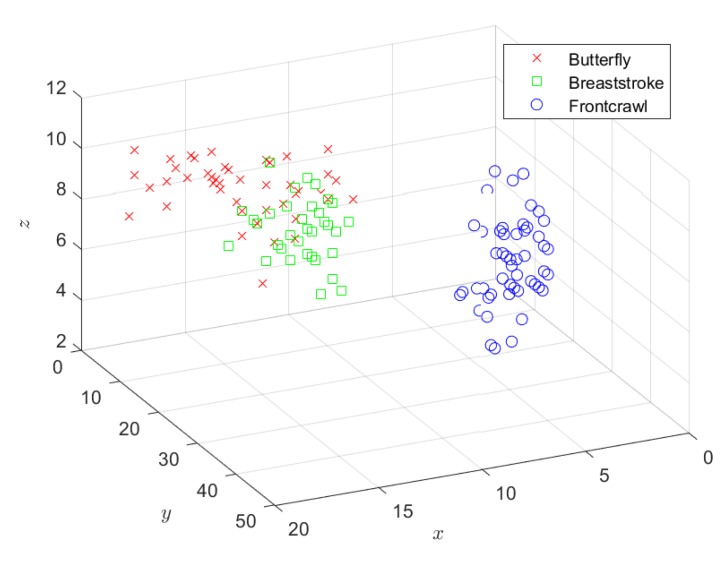
Stroke estimation dataset with the energy computed (according to Equation (2)) of each accelerometer axis (*x*, *y*, *z* in m.s^−2^).

**Table 1 sports-07-00238-t001:** Results obtained for body balance, body rotation, and trunk elevation.

Athlete Sex	Style		Body Balance (°)	Body Rotation (°)	Trunk Elevation (°)
Female	Butterfly	Min	−10.01 ± 6.87	6.17 ± 2.35	32.02 ± 7.86
		Max	−42.03 ± 3.91		
	Backstroke	Min	6.43 ± 3.00	32.98 ± 9.37	n.a.
		Max		−34.13 ± 3.44	
	Breaststroke	Min	−9.17 ± 3.60	3.31 ± 4.97	31.44 ± 6.34
		Max	−40.60 ± 4.32		
	Front Crawl	Min	−18.81 ± 0.92	49.51 ± 9.67	n.a.
		Max		−51.83 ± 9.48	
Male	Butterfly	Min	8.33 ± 9.21	−2.00 ± 3.29	38.27 ± 8.81
		Max	−29.94 ± 8.61		
	Backstroke	Min	0.01 ± 6.00	33.85 ± 15.62	n.a.
		Max		−30.13 ± 6.76	
	Breaststroke	Min	−3.59 ± 8.57	-1.48 ± 3.72	26.15 ± 8.30
		Max	−29.74 ± 7.61		
	Front Crawl	Min	−7.69 ± 5.32	47.95 ± 10.23	n.a.
		Max		−49.62 ± 9.74	

Maximum (Max) and minimum (Min) average values are presented in the dominant component of the motion in each style, while the mean value is presented in the nondominant component; n.a—not applicable.

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
