# Peer review of "SwimBIT: A Novel Approach to Stroke Analysis During Swim Training Based on Attitude and Heading Reference System (AHRS)"

_sports, 2019, doi:10.3390/sports7110238_

Round 1

Reviewer 1 Report

Thank you for your submission. I found the content of your effort sound and interesting. The comments that follow are intended to enhance your product if heeded. 

Abstract - 

I am not sure if the word 'real' needs to be used in this description. 

Perhaps a format shift to a more concise description of Purpose, Methods, Results, Conclusions. 

Introduction - 

Please expand the first paragraph or have the information absorbed elsewhere. Its brevity appears awkward. 

Please offer a definitive research question or purpose statement at the end of the introduction. 

Methods - 

Check for language continuity. 

Move Figure 1 below the Hardware section. 

Page 4 line 140 is a hanging sentence. 

Page 5 line 196 transitions oddly.

Results - 

Consider consistent spacing between text and formula throughout the paper. 

Shift text so Table 1 can be presented as one unit, not split between two pages. 

Make wise choices when using numbers (1,2,3,....) and text (one, two, three...) in the paper. To many numbers makes consuming your product choppy. 

Discussion - 

Elevate the language to be more concise. 

It seems the discussion could be expanded to provide greater translational applicability in the field. 

I believe with a bit of effort your team can craft a more refined manuscript worthy of further consideration of publication. The utility of this research is the strength and should be a consideration for other publications in the field. 

Author Response

please see the attach.

Reviewer 2 Report

Very interesting method to detect drawbacks and mistakes of an individual swimmer in all 4 stroke types. It is probably best used in crawl especially if used in triathletes. I recommend publication of this paper in order to get more people interested in this method. Every-day use will then show the value of the method on a long term basis for improvement of swimming times in competitions. 

Author Response

please see the attach

Reviewer 3 Report

The authors must be congratulated for the development of this innovative system. Definitely this automatic data collection device has powerful advantages for sport trainers and scientists, it is simple, wearable, accurate and non-pretentious (i.e., the technology used is sensitive to measure the indicated performance parameters). The study presents a solid rationale and it is well written.

Please find some minor comments below.

Title: I just googled “SwimIT” and found one different device (https://theswimit.com/) and one psychological academy for swimmers (https://www.teamswimit.com/). If this system has not been commercialized yet, I would suggest using a different name to avoid mix-ups.

L73: Please replace the º by the proper degree symbol ° throughout the manuscript.

L73: I would also suggest using a more scientific approach for the gyroscope outcome, like “deg·s-1”.

L82: Please detailed the size and weight of the device and the bag. I assume it is small and light, but thinking on the potential transfer to children, I find this information pertinent.

L171: Please add further information about differences and advantages between the automatic classifiers SVM and ANN to clarify the readers.

L231-232: Please rephrased as “Two different automatic classifiers (SVM and ANN) were tested to distinguish the other three techniques”

Table 1. I find this table difficult to read; if rows are min/max, then what means the merge rows? (e.g., Female/Butterfly/Body Rotation: 6.12 ± 2.35). Please define “n.a.” in the footer.

L268: The authors specified this is a low-cost system but I did not find any information about the prize (maybe I lost it). Please include an estimated cost to support this statement.  

L274: The authors did not show any limitations along the discussion. Certainly, these are promising results for a prototype, but somehow needs improvements that must be discussed. For instance, the fact that 1 out of 10 strokes are not correctly classified neither counted seems to be a drawback.

Author Response

please see the attach
